# The Contribution Ratio of Autotrophic and Heterotrophic Metabolism during a Mixotrophic Culture of *Chlorella sorokiniana*

**DOI:** 10.3390/ijerph18031353

**Published:** 2021-02-02

**Authors:** Jeong-Eun Park, Shan Zhang, Thi Hiep Han, Sun-Jin Hwang

**Affiliations:** 1Department of Environmental Science and Engineering, Kyung Hee University, Seochon-dong, Giheung-gu, Yongin-si 17104, Gyeonggi-do, Korea; peacefulluv1@gmail.com; 2Jining Municipal Eco-Environmental Monitoring Center, Jining 272000, Shandong, China; 15066373032@163.com; 3School of Chemical Engineering, Yeungnam University, Gyeongsan 38541, Gyeongbuk, Korea

**Keywords:** autotrophic metabolism, *Chlorella sorokiniana*, heterotrophic metabolism, mixotrophic metabolism, wastewater treatment

## Abstract

The contribution ratio of autotrophic and heterotrophic metabolism in the mixotrophic culture of *Chlorella sorokiniana* (*C. sorokiniana*) was investigated. At the early stage of mixotrophic growth (day 0–1), autotrophy contributed over 70% of the total metabolism; however, heterotrophy contributed more than autotrophy after day 1 due to the rapid increase in cell density, which had a shading effect in the photo-bioreactor. Heterotrophy continued to have a higher contribution until the available organic carbon was depleted at which point autotrophy became dominant again. Overall, the increase in algal biomass and light conditions in the photo-bioreactor are important factors in determining the contribution of autotrophy and heterotrophy during a mixotrophic culture.

## 1. Introduction

Microalgae are a promising source of alternative fuels because they perform photosynthesis by fixing atmospheric carbon dioxide and have high growth rates and areal productivity [1]. Moreover, the use of microalgae to treat wastewater has been widely studied due to their ability to remove nitrogen, phosphorus and carbon, which are essential nutrients for their growth [2]. Furthermore, aerobic wastewater treatment using microalgae requires a low energy input and decreases sludge production [3,4]. The biomass of microalgae after an industrial wastewater process possesses the possibility of converting industrial waste to new sources of materials for energy [5]. Microalgae represent promising biological systems for treating a variety of sources of wastewater due to their metabolic flexibility, i.e., their ability to perform photo-autotrophic, mixotrophic or heterotrophic metabolism [6,7].

Many algal species are not only capable of growing autotrophically, using photons of light as a sole energy source, but are also capable of growing heterotrophically, using organic carbon as an energy and carbon source for metabolism [1]. Furthermore, some microalgae can use both inorganic carbon (IC) and organic carbon (OC) simultaneously for growth, which is called mixotrophic metabolism. Among the three metabolisms, mixotrophic growth has several advantages for wastewater treatment because cell growth is independent of light, which is often a physiological limiting factor in algal culture through photo-inhibition and creates cost due to the energy requirements for the culture system. Therefore, mixotrophic cultures can reach a high final dry weight and growth rate with minimal photo-inhibition caused by high intensity light [8,9,10]. In addition, oxygen produced in part by autotrophic metabolism during a mixotrophic culture can lower the costs of ventilation. For these reasons, the use of mixotrophic microalgae for wastewater treatment systems and bio-fuel feedstock has grown in popularity for its economic and technologic aspects [2]. The most common mixotrophic cultivation is conventional photo-autotrophy supplemented with some form of organic carbon [11,12].

In terms of wastewater treatment, the mixotrophic culture is the most efficient culture because not only nitrogen and phosphorus but also organic compounds from wastewater can be removed simultaneously.

However, the culture conditions that cause a shift between autotrophy and heterotrophy and the consequent growth efficiency of mixotrophic algae are not fully understood. Thus, in order to use mixotrophic cultures to remove nutrients and organic matter from wastewater, the preferable contribution ratio of organic to inorganic carbon must be investigated.

The stoichiometries of autotrophic, heterotrophic and mixotrophic metabolism were revealed by a previous study [8], which are shown in Equations (1), (2) and (3), respectively.

Autotrophic metabolism:H_2_O + HCO_3_^−^ → C (Biomass) + 1/2O_2_ + 3OH^−^(1)

Heterotrophic metabolism:
(1+a)CH_2_O + O_2_ → C (Biomass) + aCO_2_ + (1+a) H_2_O(2)


Mixotrophic metabolism:
b HCO_3_^−^ + cCH_2_O → (b+(c−a)) C (Biomass) + 3OH^−^ + aCO_2_(3)


Generally, in autotrophic cultivation, IC consumed by algae can be calculated by measuring the initial and residual concentrations of bicarbonate. Likewise, in heterotrophic cultivation, consumed OC can be determined by estimating the difference between the initial and residual OC concentrations provided. In mixotrophic cultivation, however, IC is produced by heterotrophic respiration (Equation (2)) and is added to the residual IC concentration as an available carbon source. This is the main difficulty in defining the ratio of autotrophic to heterotrophic contributions.

Despite the significance of distinguishing between autotrophic and heterotrophic contributions in mixotrophic metabolism, this topic has been rarely studied mainly due to the complex characteristics of mixotrophy compared with autotrophy or heterotrophy alone. A study investigated the balance of autotrophy and heterotrophy in a mixotrophic growth of *Karlodinium micrum* and found that it was dominated by heterotrophic metabolism [13]. This study investigated the feasibility of applying mixotrophic microalgae cultures to a wastewater treatment. The contribution ratio of autotrophy and heterotrophy and their changes according to culture processes were specifically investigated. The contribution ratio during the mixotrophic growth of the microalgae was determined with a stoichiometric and experimental approach. *Chlorella sorokiniana* (*C. sorokiniana*) was used as a microalgae strain instead of *Chlorella vulgaris* because it was identified as an ideal candidate for mixotrophic cultivation [14]. *C. sorokiniana* also has a higher specific growth rate, biomass productivity and cell doubling time than *Chlorella vulgaris* for the post treatment of dairy wastewater plant effluent [15].

## 2. Materials and Methods

### 2.1. Microalgae Strain, Culture Medium and Cultivation

The microalgae strain *C. sorokiniana* (AG20740) was obtained from KCTC (Korean Collection for Type Cultures, Daejeon, Korea).

In order to acclimatize in a mixotrophic culture, the microalgae were pre-cultured in 250 mL Erlenmeyer flasks containing a modified BG11 medium with NaHCO_3_ 0.5 g-C L^−1^ and glucose 0.5 g-C L^−1^. The flasks were agitated at 120 rpm using an orbital shaker and controlled at 25 ± 2 °C under a continuous photosynthetic photon flux density (PPFD) of 120 μmolm^−2^s^−1^ from a light emitting diode measured by a photo-radiometer (LI-1400, LI-COR Inc., Lincoln, NE, USA). The composition of the BG11 medium was NaNO_3_ (1500), K_2_HPO_4_ (40), MgSO_4_·7H_2_O (75), CaCl_2_· 2H_2_O (36), Citric acid·H_2_O (6), Ferric ammonium citrate (6), EDTA (1), H_3_BO_3_ (2.86), MnCl_2_·4H_2_O (1.81), ZnSO_4_·7H_2_O (0.22), NaMoO_4_·2H_2_O (0.39), CuSO_4_·5H_2_O (0.079) and Co(NO_3_)_2_·6H_2_O (0.049) as mg L^−1^.

After four days of pre-culturing, the microalgae were inoculated in the 2.5 L photo-bioreactor (working volume of 2 L) with the initial inoculum at 0.1 optical density (OD). Other operating conditions are described in Table 1.

### 2.2. Analysis of Organic Carbon, Inorganic Carbon, Cell Density and Chlorophyll-a

The residual concentration of glucose was measured with a dinitrosalicylic acid (DNS) reagent [16]. The residual IC was analyzed by a total organic carbon analyzer (TOC-VCSN, Shimadzu Corp., Kyoto, Japan). To analyze chlorophyll-a, 10 mL of algal suspension was collected from the reactor and filtered through a 0.45 μm glass microfiber filter/Circles (GF/C). The filtrate was mixed with 90% acetone and stored at 4 °C in the dark for 24 h. The filtrate was then centrifuged at 3000 rpm for 20 min. The OD of the supernatant was measured at wavelengths of 664, 647 and 630 nm with a spectrophotometer (Optizen POP, Mecasys Co., Ltd., Daejeon, Korea). The concentration of chlorophyll-a in the extract was calculated using Equation (4).
C_a_ (mg L^−1^) = 11.85(OD_664_) − 1.54(OD_647_) − 0.08(OD_630_)(4)
where C_a_ was the concentration of chlorophyll˗a in the extract and OD_664_, OD_647_ and OD_630_ were the corrected optical densities at the respective wavelengths.

The amount of pigment per unit volume was calculated as follows (Equation (5)):(5)$$\mathrm{Chlorophyll}¯a\;\left({\mathrm{mg}\;m}^{-3}\right)\;=\;\frac{C_{a}\times \mathrm{extracted}\;\mathrm{volume}\;\left(L\right)}{\mathrm{volume}\;\mathrm{of}\;\mathrm{sample}\;\left(m^{3}\right)}.$$

### 2.3. Contribution Ratio of Autotrophy and Heterotrophy in the Mixotrophic Culture

The contribution ratio of autotrophy and heterotrophy during the mixotrophic culture was calculated based on the carbon consumption as below (Equations (6) and (7)):(6)$$\mathrm{Autotrophic}\;\mathrm{contribution}\;\left(\%\right)\;=\;\frac{{\mathrm{IC}}_{A}}{{\mathrm{IC}}_{A}+{\;\mathrm{OC}}_{H}}\;\times \;100$$
(7)$$\mathrm{Heterotrophic}\;\mathrm{contribution}\;\left(\%\right)\;=\;\frac{{\mathrm{OC}}_{H}}{{\mathrm{IC}}_{A}+{\;\mathrm{OC}}_{H}}\;\times \;100$$
where ${\mathrm{OC}}_{H}$ was the amount of organic carbon consumed by heterotrophic metabolism, which could be determined by estimating the difference between the initial and residual amount of OC (mg). ${\mathrm{IC}}_{A}$ was the amount of inorganic carbon consumed by autotrophic metabolism, which was calculated by (Equation (8)):(8)$${\mathrm{IC}}_{A}\;\left(\mathrm{mg}\right)\;=\;\Delta \mathrm{IC}\;+{\;\mathrm{IC}}_{H}$$
where ΔIC was the total inorganic carbon removal, which could be determined by estimating the difference between the initial and residual amounts of inorganic carbon (mg). ${\mathrm{IC}}_{H}$ was inorganic carbon produced by heterotrophic metabolism, which was calculated with the equation below (Equation (9)):(9)$${\mathrm{IC}}_{H}\left(\mathrm{mg}\right)\;={\;\mathrm{OC}}_{H}\times \;\frac{\mathrm{Theroretical}\;\mathrm{IC}\;\mathrm{production}}{{\mathrm{OC}}_{H}}.$$

As NaHCO_3_ and glucose were used as IC and OC sources, respectively, for mixotrophic metabolism, the stoichiometry equation of mixotrophic metabolism can be expressed as follows [5] (Equation (10)):HCO_3_^−^ + 6CH_2_O → 2C (Biomass) + 3OH^−^ + 5CO_2_.(10)

This equation shows that 1 mol of glucose could produce 5/6 mol of IC. Therefore, the amount of IC produced by heterotrophy was calculated theoretically. The contribution ratio was expressed as a relative abundance of each type of metabolism.

## 3. Results

### 3.1. Autotrophy and Heterotrophy Contribution in Mixotrophic Metabolism

The residual OC and IC concentrations over time under mixotrophic conditions are shown in Figure 1. OC was removed rapidly from day 1 to day 3 by heterotrophic metabolism while the residual IC concentration increased slightly due to heterotrophic respiration during that period. Afterward, autotrophic metabolism consumed the remaining IC, hence the IC concentration in the artificial wastewater decreased.

The variations in the contribution ratio of autotrophic and heterotrophic metabolism over time during mixotrophic growth are shown in Figure 2. Autotrophic metabolism contributed 73% of the total metabolism during the initial growth phase (day 0–day 1) while heterotrophic metabolism contributed relatively little at 27% of the total metabolism. After day 1, OC began to be consumed rapidly and the amount consumed increased until day 3 (Figure 1). Consequently, heterotrophic metabolism took precedence over autotrophic metabolism during day 1–day 3. As previously mentioned, residual OC was depleted on day 3–4 and autotrophic growth contributed as much as 78% of the metabolism during that period. Consequently, *C. sorokiniana* completely switched to autotrophy after day 4. Heterotrophic metabolism was expected to surpass autotrophy considering the high energy needed for Calvin cycle processes [17], which is the main pathway for auto-photosynthesis. In reverse, heterotrophic metabolism requires less energy. Therefore, we expected that heterotrophic metabolism would occupy a higher contribution in the early growth stage and would gradually change to autotrophic metabolism resulting in the consuming of OC and releasing of CO_2_. However, our results showed the opposite pattern; the dominant metabolism was autotrophic at the early stage of culture. The possible reason may be explained in the next section.

### 3.2. Variation in Chlorophyll-a Content During the Mixotrophic Culture

In order to figure out why autotrophic metabolism was dominant at the early stage of the mixotrophic culture, the chlorophyll-a content and cell density were observed during the experiment. The variations of chlorophyll-a content and cell density with time are shown in Figure 3. Cell density OD_660_ increased steadily from 0.1 to 3.3 from day 1 to day 5. In contrast, chlorophyll-a concentration increased rapidly to 14 mg g-cell^−1^ just after 1 day then became stable to day 3 and decreased gradually.

On day 0–1, the autotrophic contribution was higher due to the low initial cell density (OD_660_ 0.1) and sufficient light penetration. Therefore, photosynthesis was the main metabolism for *C. sorokiniana* in the culture system. At a low light intensity, the microalgae synthesized photosynthetic pigments to absorb photons more actively [18], resulting in a doubled increase of the chlorophyll-a concentration from day 0 to day 1. Afterwards, the cell density increased drastically and the optical density was relatively high (OD_660_ 0.7) within 24 h. This cell density might be high enough to decrease the light penetration through a photo-bioreactor, resulting in insufficient light intensity for proper photosynthesis. It has been reported that a high chlorophyll-a content and algal density (at concentrations of 10^5^ cells/mL or above) caused a self-shading effect that limited the light for photosynthesis [19,20].

The present results support the possibility of a rapid change in the dominant metabolism from autotrophy to heterotrophy at the early stage of a mixotrophic algal culture from day 1 to day 3 (Figure 2) due to inhibited light penetration by a high chlorophyll-a content and high cell density in the photo-bioreactor. Similarly, [8] explored the metabolic flux of autotrophic and heterotrophic metabolism in a mixotrophic culture of *Spirulina sp.* They suggested that mixotrophic metabolism depends on a photo-chemical reaction and *Spirulina sp.* produced the necessary ATP mainly from light reactions at the early stage of growth. The ATP generated from photosynthesis could accelerate glucose anabolism in heterotrophic metabolism thereafter. As the two metabolisms were coupled, the final algal biomass of mixotrophic conditions was much higher than in autotrophy or heterotrophy alone.

There is another possible explanation for initial dominance of autotrophy. According to [21], light can significantly affect the glucose uptake by *Chlorella vulgaris* by inhibiting the activation of the hexose/H^+^ symport system, which brings glucose into the cells. Thus, light can directly inhibit glucose uptake in heterotrophic metabolism and, consequently, can allow active autotrophic metabolism at the early stage of algal growth. In general, glucose uptake can be temporarily inhibited on day 0–1 but this inhibition can be mitigated when the cell density is increased to an OD_660_ higher than 0.7. After the light inhibition was mitigated, heterotrophic metabolism became dominant and OC was actively consumed.

This study emphasizes the role of the self-shading effect on the contribution ratio of autotrophy and heterotrophy and their changes during the mixotrophic growth of microalgae. The result infers that the self-shading effect is very important in consuming IC from the middle stage of the experiment. That finding is meaningful for the practical use of microalgae for a wastewater treatment process where OC and IC are usually present together. Different from bacterial sludge, excess microalgae sludge is high added value material that provides raw materials for bio-fuel, animal food, etc. The results presented in this study help to discover the condition at which both the removal efficiency and microalgae biomass are highest.

## 4. Conclusions

The contribution ratio of autotrophy and heterotrophy changed dramatically during a mixotrophic culture. Autotrophy dominated the growth metabolism during the early culture period (day 0–1). However, increased algal biomass might inhibit factor photosynthesis through self-shading. Heterotrophic metabolism became dominant as *C. sorokiniana* began to consume organic carbon (day 1–3). After day 3, as available organic carbon became scarce, autotrophy became completely dominant. The results suggest that increased algal biomass and consequent self-shading are important factors in determining the contribution of autotrophy and heterotrophy during a mixotrophic culture. The study contributed to clarifying the dynamic of mixotrophic metabolism, which is useful for the experts of the sector to understand algae growth.

## Figures and Tables

**Figure 1 ijerph-18-01353-f001:**
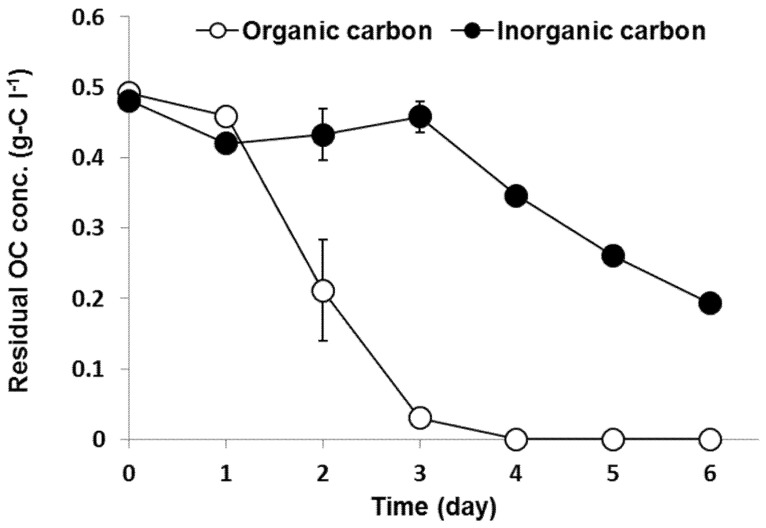
Variations in residual organic carbon (OC) and inorganic carbon concentrations over time in artificial wastewater under mixotrophic conditions. Artificial wastewater was prepared in a modified BG11 medium with NaHCO_3_ 0.5 g-C L^−1^ and glucose 0.5 g-C L^−1^ (error bars denote standard deviation (*n* = 3)).

**Figure 2 ijerph-18-01353-f002:**
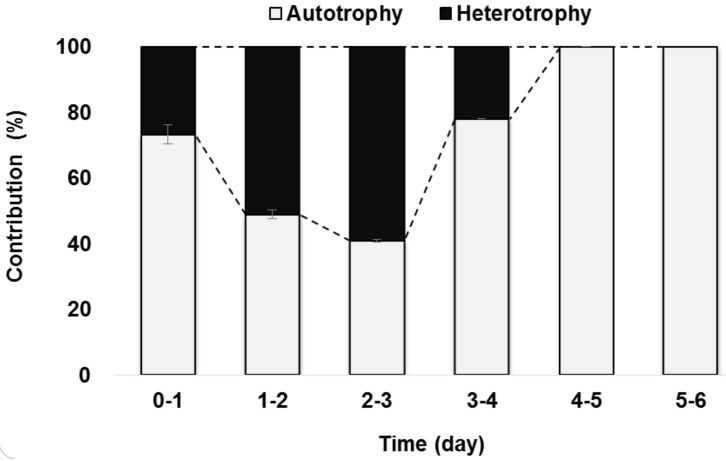
Variations in the contribution ratio of autotrophic and heterotrophic metabolism during six days of the mixotrophic culture. Artificial wastewater was prepared in a modified BG11 medium with NaHCO_3_ 0.5 g-C L^−1^ and glucose 0.5 g-C L^−1^. The contribution ratio of autotrophic and heterotrophic metabolism was calculated based on the amount of consumed NaHCO_3_ and glucose, respectively.

**Figure 3 ijerph-18-01353-f003:**
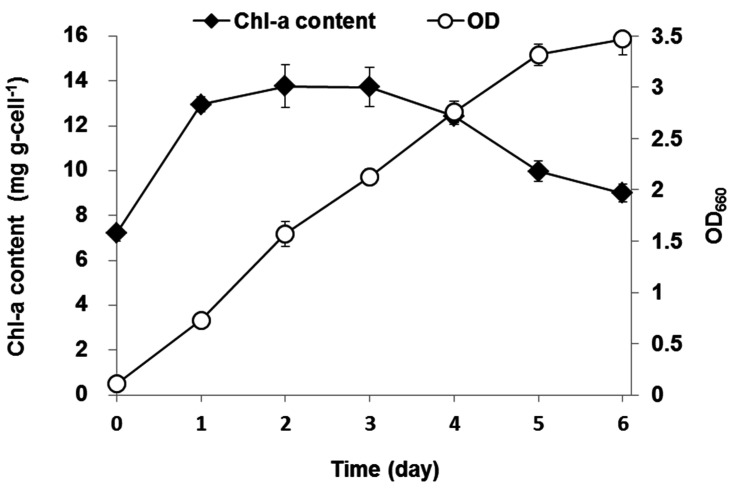
Variations in chlorophyll-a (Chl-a) content and optical density (OD_660_) over time under a mixotrophic cultivation. Artificial wastewater was prepared in a modified BG11 medium with NaHCO_3_ 0.5 g-C L^−1^ and glucose 0.5 g-C L^−1^ (error bars denote standard deviation (*n* = 3)).

**Table 1 ijerph-18-01353-t001:** Experimental conditions of a photo-bioreactor in a mixotrophic culture.

Parameter	Detail
Microalgae strain	*C. sorokiniana*
Culture type Organic carbon Inorganic carbon	Mixotrophic culture Glucose: 0.5 g-C L^−1^ Na_2_CO_3_: 0.5 g-C L^−1^
Artificial wastewater	Modified BG11 (NaNO_3_: 150 mg-N L^−1^, K_2_HPO_4_: 30 mg-P L^−1^)
pH and Temp.	8 ± 0.3 and 25 ± 2 °C
PPFD	120 μmolm^−2^s^−1^
Working volume	2 L
Mixing rate	100 rpm
Initial inoculum	0.1 OD
Light source	White LED
Light/Dark cycle	24 : 0 (h)

PPFD: photosynthetic photon flux density; OD: optical density; LED: light emitting diode.

## Data Availability

Not applicable.

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
