# Peer review of "The Contribution Ratio of Autotrophic and Heterotrophic Metabolism during a Mixotrophic Culture of Chlorella sorokiniana"

_ijerph, 2021, doi:10.3390/ijerph18031353_

Round 1

Reviewer 1 Report

The authors presented a study of the autotrophic and heterotrophic metabolism in the mixotrophic culture of Chlorella sorokiniana.

The subject of the article is relevant and of interest to the readers of the journal, however, it is recommended that the authors work more on the development of the paper since the current version does not contain enough results to support their conclusions.

It is recommended to focus mainly on the following:

In the introduction section it is mentioned that " Despite the significance of distinguishing between autotrophic and heterotrophic contributions in mixotrophic metabolism, this topic has been rarely studied, mainly due to the complex characteristics of mixotrophy compared with autotrophy or heterotrophy alone" however, a proper detailed description of the state of the art was not made

The authors mention in the title that the study refers to wastewater but in the materials and methods section it is described that synthetic water was added.

A longer-term study is required, results are only shown for 6 days of operation in the reactor.

There are only 10 articles in the references

Author Response

It is recommended to focus mainly on the following:

-In the introduction section it is mentioned that " Despite the significance of distinguishing between autotrophic and heterotrophic contributions in mixotrophic metabolism, this topic has been rarely studied, mainly due to the complex characteristics of mixotrophy compared with autotrophy or heterotrophy alone" however, a proper detailed description of the state of the art was not made.

Response:

Thank you very much for your comment. The state of the art was added in line 70-71.

-The authors mention in the title that the study refers to wastewater but in the materials and methods section it is described that synthetic water was added.

Response:

The title was revised accordingly.

-A longer-term study is required, results are only shown for 6 days of operation in the reactor.

Response:

Thank you for your comment. With the current COVID-19 pandemic and limited time, it is very difficult to perform the experiment with more substrates, we will consider that suggestion in the next study. Please understand for our situation.

-There are only 10 articles in the references

Response:

8 more references were added.

Reviewer 2 Report

Overall, the paper is well written. I would like to see more conclusive remarks (expanded "conclusion" section). There is only one sentence regarding the overall conclusions and on discussion of a potential impact or what these conclusions mean.

In addition, the paper cites only 10 references total. It would strengthen both the introduction and the conclusion to review a wider depth of current literature.

Author Response

Overall, the paper is well written. I would like to see more conclusive remarks (expanded "conclusion" section). There is only one sentence regarding the overall conclusions and on discussion of a potential impact or what these conclusions mean.

Response:

The conclusion was expanded. Please refer to line 200-202 in the manuscript.

In addition, the paper cites only 10 references total. It would strengthen both the introduction and the conclusion to review a wider depth of current literature.

Response:

The introduction was strengthened. Please refer to line 30-34, line 47-48, and line 70-71 in the revised manuscript.

The potential impact of the results to praxes was added in conclusion from line 200-202.

8 more references were added to the revision manuscript.

Reviewer 3 Report

The application of mixotrophic metabolism of various organisms for wastewater treatment constitutes a very interesting approach especially from the economical point of view. The present contribution is a part of a trend of such research.
I would recommend its publication in International Journal of Environmental Research and Public Health after considering my comment below:

The equations (1) and (10), pages 2 and 4, respectively, are quite incomprehensible for me. What does anion HCO3- mean ? (1) Should there be a negative charge in the formula HCO3 ? (10)

Author Response

The equations (1) and (10), pages 2 and 4, respectively, are quite incomprehensible for me.

What does anion HCO3- mean ? (1) Should there be a negative charge in the formula HCO3 ? (10)

Response:

Thank you very much for your comment. The HCO3 means bicarbonate ion. The formula of it was revised in equation (1) and (10)

Reviewer 4 Report

The manuscript is of interest, but it is quite week. In fact it presents only results of one experiment. I can not recommend its publication in the present form, but I encourage you to resubmit, if editors approve that.

Key issues:

  1. The introduction shall be enhanced, especially about the literature regarding micotrofic grouwth and especially its applicability in the waste-water treatment (more particular references). In this state it is very trivial and well-known.
  2. You have tested the heterotrophic metabolism on glucose only. For generalization, more substrates shall be tested.
  3. The manuscript lacks discussion about th epotential impact of the results to praxes.
  4. In such type of the work, I am significantly missing the Monod's calculations, especially the gain of biomass vers. different sources of carbon.
  5. Used statistical methods need to be specified.
  6.  I would prefer Figure 2 to be absolute.

Author Response

Key issues:

  1. The introduction shall be enhanced, especially about the literature regarding mixotrophic growth and especially its applicability in the waste-water treatment (more particular references). In this state it is very trivial and well-known.

Response:

The introduction was strengthened. Please refer to line 30-34, line 47-48, and line 70-71 in the revised manuscript.

  1. You have tested the heterotrophic metabolism on glucose only. For generalization, more substrates shall be tested.

Response:

We highly appreciate your suggestion. With the current COVID-19 pandemic and limited time, it is very difficult to perform the experiment with more substrates; we will do that in the next study. Please understand for our situation.

  1. The manuscript lacks discussion about the potential impact of the results to praxes.

Response:

The potential impact of the results to praxes was added in line 200-202.

  1. In such type of the work, I am significantly missing the Monod's calculations, especially the gain of biomass vers. different sources of carbon. Used statistical methods need to be specified. I would prefer Figure 2 to be absolute.

Response:

Thank you very much for your comment. With the current COVID-19 pandemic and limited time, it is very difficult to perform the experiment with more substrates; we will do that in the next study. Please understand for our situation.

Reviewer 5 Report

The main objective of the study is the evaluation of mixotrophic algae cultivation a complex subject that need further research achievements to be proper known. The paper try to clarify the dynamic of the algae growth, introducing added information, proposing mechanisms to justify the process performance.  

The research is adequately planned, with the right instruments and parameters to depict the growth. The data are presented in clear comprehensive graphics. The results are interesting and defines the behavior of the autotrophic and heterotrophic phases, introducing novel data. The paper is written in good English and has the right length. It is easy to read and is not boring, excluding a portion of discussion, which can be improved. Globally the paper is interesting for the expert of the sector as try to explain the mechanisms underlying the algae growths. The paper has good quality and is suitable to be published as it is.

Author Response

Response:

Thank you very much for your kind comments and acceptance to publish our manuscript in this journal.

Round 2

Reviewer 1 Report

There is an improvement in the article after the first version.
I still consider that an evaluation was made with little experimental data, however, due to the prevailing situation I would like to suggest that the authors extend the discussion of the results further in order to consider their publication.
The use of the microalgae Chlorella sorokiniana needs to be justified, since almost all the literature analyzed includes C. vulgaris.
The paragraph in lines 166-168 should be reviewed in its wording.It is also necessary to validate the amount of chlorophyll at which it causes an effect on light by attaching other previous studies.

Author Response

I still consider that an evaluation was made with little experimental data, however, due to the prevailing situation I would like to suggest that the authors extend the discussion of the results further in order to consider their publication.

The use of the microalgae Chlorella sorokiniana needs to be justified, since almost all the literature analyzed includes C. vulgaris.

Response:

The use of the microalgae Chlorella sorokiniana was justified in introduction part from line 75-79:

“Chlorella sorokiniana was used as microalgae strain instead of Chlorella vulgaris because it is identified as an ideal candidate for mixotrophic cultivation [14]. Especially, Chlorella sorokiniana has higher the specific growth rate, the biomass productivity, and the cell doubling time than Chlorella vulgaris for post treatment of dairy wastewater treatment plant effluent [15].”

  1. Li, T.; Zheng, Y.; Yu, L.; Chen, S. Mixotrophic cultivation of a Chlorella sorokiniana strain for enhanced biomass and lipid production. Biomass and Bioenergy. 2014, 66, 204–13.
  2. Asadi, P.; Rad, H.A.; Qaderi, F. Comparison of Chlorella vulgarisand Chlorella sorokiniana in post treatment of dairy wastewater treatment plant effluents. Environ. Sci. Pollut. Res2019, 26, 29473–29489.

The paragraph in lines 166-168 should be reviewed in its wording. It is also necessary to validate the amount of chlorophyll at which it causes an effect on light by attaching other previous studies.

Response:

The paragraph was revised and highlighted in yellow in the revised MS from line 167-169.

We can’t find the exact amount of chlorophyll a at which it causes an effect on light. However, we can find that the self-shading was found to occur only at concentrations of 105 cells/ml or above. This content was added into the revised manuscript from line 172-174.

Ref: Dorling, M.; McAuley, P.J.; Hodge, H. Effect of pH on growth and carbon metabolism of maltose-releasing Chlorella (Chlorophyta). Eur. J. Phycol. 1997, 32, 19–24.

Reviewer 4 Report

I agree with enhancing of the Introduction as well as obstacles in carrying out further experiments on different sources of C.

However I can not agree with enhancing the discussion by two lines in Conclusion. Potential readers are also people from praxis and they would like to read how difficult is e.g. the upscale to real waste-water treatment plant, whether the technology is potentially feasable. I really suggest to enhance this part (seprate section would be interesting). Also the Conclusions should not state anything not previously discussed.

The Monod's parameters do not require further experiments, it is just calculations. At least for the glucose you should be able to calculate them from data (an summarize e.g. in the table) and also compare with other published parameters. Yield of biomass from a substrate and other Monod's parameters are very valuable in modeling of the technology parameters.

The statistical methods are still not defined, I can suggest adding a new section to Materials and methods. Note that MDPI is an open-access publisher not limiting the length of the manuscript. You should not explain the basis of the methods, but summarizing used tests, software, confidence levels etc. is important.

Author Response

1. I agree with enhancing of the Introduction as well as obstacles in carrying out further experiments on different sources of C.

However I cannot agree with enhancing the discussion by two lines in Conclusion. Potential readers are also people from praxis and they would like to read how difficult is e.g. the upscale to real waste-water treatment plant, whether the technology is potentially feasible. I really suggest to enhance this part (separate section would be interesting). Also the Conclusions should not state anything not previously discussed.

Response:

Thank you for your suggestion. The discussion part was enhanced from line 197-204 in the revised manuscript.

“This study emphasizes the role of self-shading effect on contribution ratio of autotrophy and heterotrophy and their changes during mixotrophic growth of microalgae. The result infers that the self-shading effect is very important in consumming IC from the middle stage of the experiment. That finding is very important for practical use of microalgae for wastewater treatment process where OC and IC are usually present together. Different with bacterial sludge, excess microalgae sludge is high added value material which provides raw materials for bio-fuel, animal-food, etc... The results presented in this study helps to find out the condition at which both the removal efficiency and microalgae biomass are highest. “

2. The Monod's parameters do not require further experiments, it is just calculations. At least for the glucose you should be able to calculate them from data (an summarize e.g. in the table) and also compare with other published parameters. Yield of biomass from a substrate and other Monod's parameters are very valuable in modeling of the technology parameters.

Response:

We absolutely agree with the reviewer that Monod model parameters are highly useful for readers who are interested in modeling for our data for further and future field applications. One thing we would like to note, however, that we collected our experimental data from a single 2.5 L bioreactor and monitored carbon consumption and microalgal growth (yield). 

Upon the reviewer’s suggestion for Monod model parameters, we had to conduct a separate set of kinetic experiments: e.g., An independent set of microbial growth kinetic experiments as a function of initial substrate (carbon source) concentration (various substrate concentrations) must be conducted and the growth rates must be analyzed separately at each initial substrate concentration. And then, such data (nonlinear behavior of growth rate as a function of substrate) needs to be analyzed by the concept of the Monod model. 

Unfortunately, we did not perform these separate kinetic experiments, and this is why we don’t think that we are able to collect the Monod parameter from our current data set as the reviewer suggested. Also, it would require a considerable amount of additional experiments if we would try to collect the Monod parameters for different sources of carbon (as suggested by the reviewer in the first stage of review) because additional independent kinetic experiments are subjected to each type of carbon sources. 

We are afraid that this would go way beyond the scope of our study at this time. We still agree with the reviewer that the Monod information is very useful, so that we will include that in a continuous study we are currently performing. But we would sincerely like the reviewer to understand our original goal of this study that we focused on mixotrophic growth of microalgae based on mixed carbon wastewaters in a single bioreactor.

3. The statistical methods are still not defined, I can suggest adding a new section to Materials and methods. Note that MDPI is an open-access publisher not limiting the length of the manuscript. You should not explain the basis of the methods, but summarizing used tests, software, confidence levels etc. is important

Response:

As a matter of fact, there was no need to use any statistical software for our experimental data we collected. We did neither conduct any modeling nor statistical analysis of our data. But, instead, we are aware that we added error bars in Figure 1 and 3, and therefore, we realized that it would be better to state this clearer. Thus, we revised the figure titles accordingly (see the titles of Figure 1 and 3)